# Placental Pathology as a Tool to Identify Women for Postpartum Cardiovascular Risk Screening following Preeclampsia: A Preliminary Investigation

**DOI:** 10.3390/jcm11061576

**Published:** 2022-03-13

**Authors:** Samantha J. Benton, Erika E. Mery, David Grynspan, Laura M. Gaudet, Graeme N. Smith, Shannon A. Bainbridge

**Affiliations:** 1Department of Health Sciences, Carleton University, Ottawa, ON K1S 5B6, Canada; samanthabenton@cunet.carleton.ca; 2School of Interdisciplinary Health Sciences, University of Ottawa, Ottawa, ON K1H 8L1, Canada; emery103@uottawa.ca; 3Department of Pathology and Laboratory Medicine, University of British Columbia, Vancouver, BC V6T 1Z7, Canada; david.grynspan@interiorhealth.ca; 4Department of Obstetrics and Gynaecology, Queen’s University, Kingston, ON K7L 2V7, Canada; laura.gaudet@kingstonhsc.ca (L.M.G.); gns@queensu.ca (G.N.S.)

**Keywords:** preeclampsia, placenta, histopathology, cardiovascular disease, cardiovascular risk, postpartum screening

## Abstract

Preeclampsia (PE) is associated with an increased risk of cardiovascular disease (CVD) in later life. Postpartum cardiovascular risk screening could identify patients who would benefit most from early intervention and lifestyle modification. However, there are no readily available methods to identify these high-risk women. We propose that placental lesions may be useful in this regard. Here, we determine the association between placental lesions and lifetime CVD risk assessed 6 months following PE. Placentas from 85 PE women were evaluated for histopathological lesions. At 6 months postpartum, a lifetime cardiovascular risk score was calculated. Placental lesions were compared between CVD risk groups and the association was assessed using odds ratios. Multivariable logistic regression was used to develop prediction models for CVD risk with placental pathology. Placentas from high-risk women had more severe lesions of maternal vascular malperfusion (MVM) and resulted in a 3-fold increased risk of screening as high-risk for CVD (OR 3.10 (1.20–7.92)) compared to women without these lesions. MVM lesion severity was moderately predictive of high-risk screening (AUC 0.63 (0.51, 0.75); sensitivity 71.8% (54.6, 84.4); specificity 54.7% (41.5, 67.3)). When clinical parameters were added, the model’s predictive performance improved (AUC 0.73 (0.62, 0.84); sensitivity 78.4% (65.4, 87.5); specificity 51.6% (34.8, 68.0)). The results suggest that placenta pathology may provide a unique modality to identify women for cardiovascular screening.

## 1. Introduction

Preeclampsia (PE) is a life-threatening hypertensive disorder of pregnancy, affecting 5–7% of pregnancies worldwide [1]. Importantly, PE is a significant risk indicator for cardiovascular disease (CVD) in later life. Women diagnosed with PE have a ~4-fold increased risk of hypertension and a ~2-fold increased risk of ischemic heart disease and stroke compared to women with uncomplicated pregnancies [2,3,4,5]. Moreover, evidence suggests that women who develop severe PE during pregnancy are at the highest risk of these outcomes [4,6,7,8]. Alarmingly, studies show that the onset of CVD and CVD-related death occur at much younger ages than the general female population [6,7,8,9]. The link(s) between PE and cardiovascular risk are not fully understood, but PE may indicate the presence of underlying, often undiagnosed, cardiovascular risk factors (CVRs) [10,11]. Moreover, underlying CVRs may directly contribute to placental dysfunction associated with PE; however, this relationship has yet to be fully elucidated [12,13].

Histopathological lesions of maternal vascular malperfusion (MVM) are commonly observed in placentas from PE pregnancies, particularly in cases of severe, early-onset disease [14,15,16]. These lesions, including increased syncytial knots and accelerated villous maturation, are believed to reflect placental hypoxia and oxidative stress arising from incomplete uterine artery modeling and abnormal uteroplacental blood flow [1,17,18]. Although common, MVM lesions are not observed in all PE cases, and a proportion of PE placentas are histologically normal or exhibit other pathology such as chronic inflammation [19,20]. Recent population-based studies have shown an association between placental lesions and future maternal health [21,22,23,24,25,26]. Catov et al. observed altered cardiometabolic profiles in women with preterm birth and lesions of MVM compared to women with term deliveries [21,26]. Additionally, women with preterm birth and co-morbid placental pathologies (MVM, inflammatory lesions) had the most severe atherogenic profiles [21]. More recently, Catov et al. demonstrated that MVM lesions are associated with vascular impairments 8–10 years after pregnancy [26]. While the mechanisms underlying these associations have yet to be fully elucidated, collectively, these studies provide strong evidence that the placenta and its pathology may provide a snapshot into future maternal cardio-metabolic health [26,27].

To reduce the burden of CVD on PE women, specialized postpartum cardiovascular health clinics are being established across North America to screen women for CVRs and initiate postpartum CVD prevention, including pharmaceutical and lifestyle interventions [28,29,30]. However, these clinics are resource-intensive, and, thus, follow-up of all PE women is prohibitive. Moreover, a proportion of PE women will remain at low risk for CVD postpartum and not require follow-up. As placental pathology is inexpensive and readily available, it may offer a unique modality to identify PE women for CVR screening. Here, we investigate the association between placental pathology and lifetime CVD risk in postpartum women following PE.

## 2. Materials and Methods

In this study, a cohort of women diagnosed with PE who underwent cardiovascular health screening at 6 months postpartum was assembled from two study sites (Kingston and Ottawa, ON, Canada).

### 2.1. Recruitment at the Kingston Site

In Kingston, women who develop a pregnancy complication are referred to the Maternal Health Clinic (Kingston General Hospital, Kingston, ON, Canada) for postpartum cardiovascular health screening at 6 months postpartum as part of routine standard of care. Assessments and evaluations, including the calculation of a lifetime cardiovascular risk score conducted at the Maternal Health Clinic, have been previously described [29]. From this clinic, we recruited eligible study participants at the time of their 6-month postpartum visit (between 2011 and 2017). Women diagnosed with PE who had placenta pathology performed at delivery of the index pregnancy (PE diagnosis) were approached to participate in the study.

### 2.2. Recruitment at the Ottawa Site

In Ottawa, women were recruited to participate in the study as part of the DREAM Study research protocol designed to emulate the Maternal Health Clinic. Women diagnosed with PE prior to delivery were recruited from inpatient services at the Ottawa Hospital General Campus (Ottawa, ON, Canada) between 2013 and 2018. Placentas from each participant were sent to Anatomical Pathology at the Children’s Hospital of Eastern Ontario (Ottawa, ON, Canada). Six months after delivery, participants returned to the Ottawa Hospital for cardiovascular health screening performed by the research study nurse. The assessments performed at this study visit were identical to those performed at the Maternal Health Clinic, and a lifetime cardiovascular risk score was calculated for each participant, as described previously [29]. At both sites, all women provided written informed consent to participate in the study.

### 2.3. Inclusion and Exclusion Criteria

PE was defined according to the contemporaneous Society of Obstetricians and Gynaecologists of Canada criteria, including hypertension (blood pressure ≥140/90 mmHg, on at least two occasions >15 min apart after 20 weeks’ gestation) with new proteinuria (≥0.3 g/day by 24 h urine collection, ≥30 mg/mmol by protein:creatinine ratio, or ≥1+ by urinary dipstick) or one or more adverse conditions (e.g., headache/visual symptoms, chest pain/dyspnea, nausea or vomiting, right upper quadrant pain, elevated WBC count) or one or more severe complications (e.g., eclampsia, uncontrolled severe hypertension, platelet count <50 × 10^9^/L, acute kidney injury) [31]. Women with chronic hypertension, known CVD prior to pregnancy, known kidney disease prior to pregnancy, or diabetes (type I, type II, gestational) were excluded. Small-for-gestational age (SGA) status was used as a proxy for fetal growth restriction and was defined conservatively as infant birth weight <5th percentile for gestational age at delivery and sex [32]. Clinical data, including medical and family history, pregnancy outcome, and postpartum cardiovascular health results, were collected by chart review following 6 months postpartum cardiovascular screening.

### 2.4. Placenta Pathology

For study participants in Ottawa, placentas were collected at the time of delivery and sent to the Pathology department. Trimmed placental weight was recorded, and gross pathology was recorded by an experienced pathology assistant. Four full-thickness tissue biopsies were randomly excised from each quadrant of the placenta, between the cord insertion site and the placental margins. Areas of visible pathology were sampled separately and not used for the full-thickness sections. Tissue was fixed in 4% neutral buffered formalin and paraffin-embedded. Following sectioning (5 μm), the tissue was stained with hematoxylin and eosin (H&E) using standard protocol [33] and stored for examination. In Kingston, archived H&E-stained tissue slides (4–5 slides/participant) were accessed from the Pathology archives for each participant. Sampling procedures were similar to those followed in Ottawa in that full-thickness biopsies were excised from each quadrant of the placenta, between the margin and cord insertion site.

At both study sites, trimmed placenta weight and gross pathology were collected from the accompanying placental pathology reports. A single, experienced placental pathologist examined the stained slides from all study participants, blinded to all clinical information apart from gestational age at delivery. Placental lesions were evaluated according to a standardized placental pathology data collection form, with pre-specified severity criteria derived from clinical practice guidelines and published literature [34]. Within the evaluation scheme, each lesion has a severity score to achieve a quantitative output for the severity of pathology. Lesions were either given a binary score of 0 (absent) or 1 (present) or a graded score according to a linear scale (i.e., 0 = absent, 1 = focal, 2 = patchy, 3 = diffuse). Individual lesions are grouped according to broad etiological categories, with a maximum severity score calculated for each category. Lesion categories and maximum severity score for each category are as follows: MVM (max score 14), implantation site abnormalities (max score 4), histological chorioamnionitis (max score 11), placental villous maldevelopment (max score 5), fetal vascular malperfusion (max score 6), maternal-fetal interface disturbance (max score 5), and chronic inflammation (max score 6). Gross anatomy (e.g., placental weight, umbilical cord length) was obtained from the corresponding historical placental pathology reports, in addition to several microscopic lesions (e.g., placental infarction, chronic deciduitis), as the tissue biopsies were collected from areas that appeared grossly normal and only included villous parenchyma (i.e., maternal decidua was not sampled).

### 2.5. Cardiovascular Risk Assessment

At 6 months postpartum, all women underwent physical and biochemical CVR screening in the Maternal Health Clinic (Kingston General Hospital) or at the Ottawa Hospital (Ottawa) according to published protocol [29]. A full medical history was taken and included information on family history, breastfeeding, and lifestyle such as smoking status and alcohol intake. A physical examination, specifically focusing on cardiovascular-related clinical predictors, was performed and collected information on weight, height, and blood pressure. A maternal blood sample was collected, and total cholesterol, LDL cholesterol, HDL cholesterol, triglycerides, glucose, and high sensitivity C-reactive protein were quantified for each participant. Physical and biochemical CVR findings were integrated to calculate a lifetime risk score for CVD, according to the previously published protocol [29]. Calculations for lifetime cardiovascular risk are based on the following risk factors: total cholesterol, systolic blood pressure, diastolic blood pressure, use of anti-hypertensive medication(s), fasting blood glucose, diagnosis of diabetes, and current smoking status. Risk stratification for each risk factor was based on predetermined measurement thresholds (optimal, not optimal, elevated, major). Lifetime cardiovascular risk estimates are also categorical and based on the total number of optimal, not optimal, elevated, and major risk factors of each individual (8%, all risk factors are optimal; 27%, ≥1 risk factor is not optimal; 39%, ≥1 risk factor is elevated; 39%, 1 risk factor is major; 50%, ≥2 risk factors are major). Lifetime cardiovascular risk estimates were simplified to categorize the women as low risk (<39% risk) or high risk (≥39% risk) for lifetime CVD. This threshold corresponds to the baseline lifetime CVD risk attributed to healthy women enrolled in the Framingham Heart Study [35].

### 2.6. Statistical Analysis

Data were analyzed using SPSS 26.0 (SPSS Inc., Chicago, IL, USA). Descriptive data were expressed as means and standard deviations for normally distributed data or medians with interquartile ranges for non-normally distributed data. Chi-square tests were used for the comparison of categorical variables, while Student’s *t*-test or Mann–Whitney U-tests were used for continuous variables. Placental lesions (frequencies and severity scores) were compared between the low CVD risk and high CVD risk groups. The association between individual placental lesions and lifetime CVD risk was assessed using odds ratios (OR) with 95% confidence intervals. Multivariable logistic regression was used to develop prediction models for lifetime CVD risk with placental data alone or in combination with clinical data. The performance of the models was assessed using area under the receiver operator characteristic (AUC ROC) curve analysis. Statistical significance was defined as *p* < 0.05.

## 3. Results

### 3.1. Clinical Characteristics

A total of 85 women were included in this study (35 from Ottawa and 50 from Kingston). The clinical characteristics of the participants, as a combined cohort and by each study site, are shown in Table 1 and Table 2. The demographic characteristics of the women between the two study sites were not significantly different, apart from maternal age and pre-pregnancy BMI. Although the average age of women in Kingston was 2 years younger than the women in Ottawa (*p* = 0.015), the mean age of participants at both sites was <34 years (31.9 ± 6.0 vs. 33.9 ± 5.6), and this was not deemed to be of high clinical relevance. Although women in Ottawa had significantly higher pre-pregnancy BMIs than women in Kingston (*p* = 0.024), the gestational weight gain for the index pregnancy was similar between the two sites (13.2 ± 7.1 vs. 14.6 ± 7.1, *p* = 0.393).

As for pregnancy outcomes, women in Kingston had significantly higher blood pressures at delivery, increased use of anti-hypertensive medication in pregnancy, and delivered at earlier gestational ages compared to women in Ottawa (34.0 (31.0, 38.0) weeks vs. 37.5 (34.4, 39.4), *p* < 0.0001) and had more SGA infants compared with women in Ottawa. At 6 months postpartum, there were no significant differences in cardiometabolic parameters between the participants at the study sites. Of the 85 women included in the analysis, 53 (62.4%) women screened high-risk for lifetime CVD at 6 months postpartum.

### 3.2. Histopathology Findings in Low- and High-Risk Women

The frequency of placenta lesions between women who screened as low- and high-risk for lifetime CVD are shown in Table 3. A total of 5 placentas (15.6%) in the low-risk group and 6 placentas (11.3%) in the high-risk group had no observed pathology (*p* = 0.74). The mean cumulative severity score (sum of scores for all categories) for the low-risk group was 3.1 ± 2.2 and 3.6 ± 2.4 for the high-risk group (*p* = 0.374). By etiological category, women who screened as high-risk for lifetime CVD had placentas with more severe lesions of MVM (score ≥ 2: 54.7% vs. 28.1%, *p* = 0.017); however, the frequency of individual lesions belonging to the MVM category was not found to be significantly different between the groups. There were no differences in the frequencies of placental lesions between high and low-risk groups when stratified by individual study site (data not shown).

### 3.3. Association of Placental Lesions and CVD Risk

Individually, no placental lesions were found to be significantly associated with high-risk CVD screening at 6 months postpartum. However, PE women with lesions of MVM with a severity score of 2 or more were more likely to screen high-risk for lifetime CVD than PE women without severe MVM lesions (severity score <2) (OR 3.10 (1.20–7.92)). Clinical data in the absence of placenta pathology findings (maternal age, gestational weight gain, blood pressure at delivery, gestational age at delivery) was moderately predictive of high-risk screening at 6 months postpartum AUC 0.68 (0.55, 0.81); sensitivity: 86.5% (74.7, 93.3); specificity: 29.0% (16.1, 46.6)) (Figure 1a). Severity of MVM lesions alone (score 2 or more) was similarly predictive of high-risk screening at 6 months postpartum (AUC 0.63 (0.51, 0.75); sensitivity: 71.8% (54.6, 84.4); specificity: 54.7% (41.5, 67.3)) (Figure 1b). However, when clinical data and MVM lesion severity were combined, the model’s predictive performance improved (AUC 0.73 (0.62, 0.84) sensitivity 78.4% (65.4, 87.5); specificity 51.6% (34.8, 68.0)) (Figure 1c).

## 4. Discussion

### 4.1. Main Findings

In this study, we observed an increase in placental histopathological lesions in women who screened high risk for lifetime CVD at 6 months postpartum following a PE pregnancy. Specifically, high-risk women had more severe lesions of MVM, the placental pathology traditionally associated with PE. MVM lesions with a severity score >2 resulted in a 3-fold increased risk of screening high risk for lifetime CVD at 6 months postpartum. The cumulative severity of MVM lesions was important in this association, suggesting that there may be critical thresholds of placental damage that reflect increased lifetime cardiovascular risk.

### 4.2. Interpretation

Previous studies have demonstrated an association between placental pathology and increased postpartum maternal cardiovascular health risk [21,22,23,24,25,26]. One study found that in normotensive women who experienced placental abruption during pregnancy, CVRs were significantly altered 6–9 months postpartum compared to women without uncomplicated pregnancies [25]. Lesions of maternal vascular maldevelopment (defined as mural hyperplasia, unaltered decidual vessels, and decidual atherosis) are associated with maternal hypertension 7 to 15 years after pregnancy [36]. Catov et al. reported that in normotensive women who delivered preterm without fetal growth restriction, those who had placental lesions of MVM and inflammation had significantly elevated atherogenic profiles assessed 4–12 years after delivery [21]. Our findings are in line with this study in which the cumulative severity of placental lesions may be important for identifying women at the highest cardiovascular risk following pregnancy. Together with our current findings, significant placental pathology may be indicative of a greater risk for CVD postpartum.

The mechanisms linking PE and postpartum maternal cardiovascular risk have yet to be fully elucidated. The most commonly held hypothesis to explain this link proposes that pre-pregnancy maternal CVRs, including both clinically diagnosed and subclinical risk factors, may contribute to the development of PE, including abnormal placental development, and predispose women to CVD after pregnancy. Placentation requires the invasion of fetal trophoblast cells into the maternal decidua, resulting in the conversion of the maternal uterine spiral arteries to high capacity, venous-like conduits to increase blood flow into the uteroplacental unit to support fetal growth and development [37]. This physiological remodeling of the uterine spiral arteries is known to be defective in PE, and the two-stage model of pathogenesis proposes that this failed remodeling leads to placental ischemia, oxidative stress, and placental dysfunction, which stimulates the release of angiogenic factors, pro-inflammatory cytokines, syncytiotrophoblast vesicles, and other factors from the placenta [1]. These processes result in characteristic histopathological lesions observed in placentas from pregnancies complicated by PE, particularly lesions of MVM [38]. Placenta-derived circulating factors interact with the maternal endothelium at the systemic level, leading to the end-organ dysfunction observed in the clinical manifestation of the disorder. The maternal environment, including subclinical CVRs common to PE and CVD, may directly contribute to impaired trophoblast invasion and defective spiral artery conversion and its downstream effects. Dyslipidemia, including elevated pre-pregnancy levels of serum triglycerides, cholesterol, LDL, and non-HDL cholesterol, have been associated with increased risk of developing PE and are known contributors to CVD [39]. Studies have shown that lipids modulate human trophoblast invasion, and alterations in maternal lipid profiles could potentially contribute to abnormal trophoblast invasion and spiral artery remodeling [40,41]. Systemic (often subclinical) inflammation, common in obesity and other cardiometabolic conditions, may also play a role in limited trophoblast invasion and spiral artery conversion during placentation in PE [42,43]. Pro-inflammatory cytokines are known to inhibit trophoblast invasion by increasing apoptosis and decreasing proliferation [44]. Cytokine imbalance prior to pregnancy may alter the maternal inflammatory milieu over and above the physiological immune/inflammatory changes that occur in pregnancy; however, exactly how this imbalance contributes to altered placentation is unknown.

PE and placental dysfunction, reflected as MVM lesions in the placenta, may also cause lasting damage to the maternal cardiovascular system that results in altered cardiovascular health trajectories. Circulating levels of inflammatory cytokines such as tumor necrosis factor-a (TNF-a) and interleukin-6 (IL-6) are elevated in PE and interfere with the maternal endothelium to produce systemic endothelial dysfunction. Women diagnosed with PE have been found to have chronically altered circulating levels of these cytokines >20 years after pregnancy, suggesting that PE may program the maternal cardiovasculature such that there is persistent cardiovascular dysregulation, precipitating CVD in later life [45]. Other maternal cardiometabolic pathways, including the renin–angiotensin system, may also contribute to abnormal placentation and reduced uteroplacental exchange, the two main features of PE [46]. While yet to be fully elucidated, dysregulation of these cardio-regulatory pathways in the mother after pregnancy may contribute to the increased risk for CVD after PE [47]. Additionally, anti-angiogenic imbalance in the maternal circulation, including elevated soluble Fms-like tyrosine kinase-1 (sFlt-1) and reduced placental growth factor (PlGF), may contribute to lasting cardiovascular dysfunction after PE. Alternations in circulating angiogenic factors during pregnancy are associated with cardiovascular changes, including increased blood pressure 6 to 12 years after pregnancy [48,49]. Although angiogenic factor levels significantly drop following delivery, a recent study suggests that angiogenic imbalance may be persistent in the postpartum period [50]. While the mechanisms of angiogenic factors on cardiovascular health are not fully known, sFlt-1 and PlGF have been shown to influence vasodilatory function in preclinical models [51,52].

It is plausible that a combination of the pre-pregnancy maternal environment and persistent alterations to the maternal cardiovascular system from placental dysfunction contributes to future CVD risk. Placental pathology, particularly lesions of MVM, may reflect both abnormalities in the maternal milieu as well as the significant cardiovascular burden from abnormal placentation, thereby identifying patients at particularly increased risk of postpartum CVD. As such, placental lesions identified at the time of delivery could provide a modality to triage PE women for cardiovascular health screening postpartum. Future studies are required to confirm the utility of placental pathology in this capacity; it may offer a unique opportunity to extend the clinical benefits of the placenta pathology exam while targeting postpartum resource-intensive cardiovascular screening to the most vulnerable patients.

### 4.3. Strengths and Limitations

The limitations of our study need to be considered. First, we did observe a significant difference in several important pregnancy parameters between our study sites, including blood pressure at delivery, anti-hypertensive use at delivery, and gestational age at delivery—indicative of a more severe form of maternal disease in our Kingston cohort. We believe these differences are likely the result of sampling bias introduced by the different study designs used across the two sites. In Kingston, a retrospective analysis was performed on placentas originally submitted to pathology based on the clinical judgment of the treating physician. Despite a PE diagnosis being identified as an indicator for placenta pathology submission, only 53% of placentas were submitted for evaluation at this site during the study period (unpublished internal audit data), in line with previous literature on the topic of suboptimal placenta pathology submission practices [53]. On the other hand, a prospective study design was employed at the Ottawa site, with 100% placental pathology submission for recruited PE study participants, irrespective of the severity of their clinical characteristics. As cardiovascular parameters were similar between our cohorts at the 6-month postpartum clinic visit, we do not feel these delivery parameters influenced our findings. Due to our small sample size, we may be underpowered to detect differences between low and high-risk groups for less common pathology, such as chronic inflammation. However, for MVM lesions, we found that a sample size of 48 was needed to detect differences between MVM scores greater than 2 within the high- and low-risk groups at the confidence level of 95% and power of 80%. Confirming our results in an adequately powered prospective study may identify a predictive combination of placental lesions that robustly identifies high-priority women for postpartum CVD screening at the time of delivery. Our study is strengthened by the standardized placental evaluations we conducted using our evidence-based synoptic data collection [34]. Variability in placenta pathology examination is a known challenge, and the use of this synoptic collection form ensures each placenta was evaluated in a reproducible manner. Moreover, our cohort study design reflects routine clinical practice in which only placentas from complicated pregnancies are submitted for histopathological examination. While placentas from uncomplicated pregnancies exhibit placental lesions with varying degrees of prevalence and severity [54], these placentas are not routinely sent for examination and the use of placental pathology in this population is limited.

## 5. Conclusions

Women with PE and severe lesions of MVM are more likely to screen as high-risk for lifetime CVD at 6 months postpartum compared to women without these lesions. Placenta pathology may provide a unique modality to identify women for postpartum cardiovascular screening. Triaging at the time of delivery would allow for targeted screening and resource allocation to the truly at-risk PE women.

## Figures and Tables

**Figure 1 jcm-11-01576-f001:**
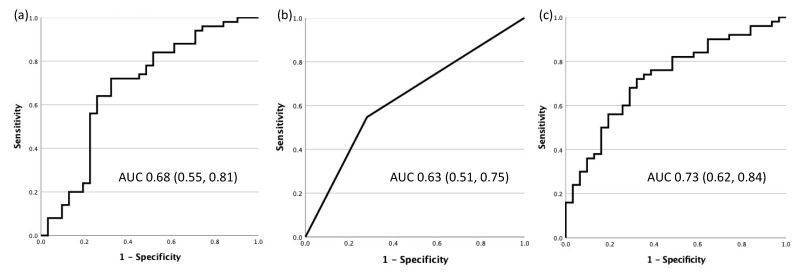
Area under the receiver operator characteristic curves for the prediction of screening high-risk for lifetime cardiovascular disease at 6 months postpartum by (**a**) maternal age, gestational weight gain, blood pressure at delivery, and gestational age at delivery, (**b**) maternal vascular malperfusion severity score, and (**c**) maternal vascular malperfusion severity score, maternal age, gestational weight gain, blood pressure at delivery, and gestational age at delivery.

**Table 1 jcm-11-01576-t001:** Maternal characteristics of the study participants as a combined cohort and by individual study site.

	Combined (*n* = 85)	Ottawa (*n* = 35)	Kingston (*n* = 50)	*p*-Value ^§^ (*t*-Test, KW, X^2^)
Maternal Characteristics				
Maternal age at delivery (y)	31.9 ± 6.0	33.9 ± 5.6	30.7 ± 5.9	**0.015**
Postsecondary education (%)	74 (88.1) ^a^	31 (91.2) ^a^	43 (86.0)	0.520
Married or common law	80 (95.2) ^a^	35 (100)	45 (91.8) ^a^	0.137
Nulliparous (%)	59 (69.4)	22 (62.9)	37 (74.0)	0.341
Pre-pregnancy BMI	24.5 (22.1, 31.0)	28.2 (23.0, 35.5)	24.4 (21.9, 28.5)	**0.024**
Smoking (%)	6 (7.1)	1 (2.9)	5 (10.0)	0.393
Previous history of HDPs (%)	11 (12.9)	7 (20.0)	4 (8.0)	0.187
Family history of CVD * (%)	44 (52.4) ^a^	16 (47.1) ^a^	28 (56.0)	0.506
Family history of PE (%)	12 (15.0) (*n* = 80)	6 (20.0) (*n* = 30)	6 (12.0)	0.520

Data presented as mean ± SD, median (IQR) or *n* (%). BMI: body mass index; CVD: cardiovascular disease; HDP: hypertensive disorders of pregnancy; IQR: interquartile range; PE: preeclampsia; SD: standard deviation. ^§^ Comparison between participants in Ottawa and Kingston. * Includes coronary artery disease and cerebral vascular disease. ^a^ Data missing for one participant.

**Table 2 jcm-11-01576-t002:** Delivery and postpartum characteristics of the study participants as a combined cohort and by individual study site.

	Combined (*n* = 85)	Ottawa (*n* = 35)	Kingston (*n* = 50)	*p*-Value ^§^ (*t*-Test, KW, X^2^)
At delivery				
Systolic BP *(mmHg)	152 ± 25	136 ± 17	164 ± 22	**<0.0001**
Diastolic BP *(mmHg)	93 ± 13	85 ± 10	98 ± 13	**<0.0001**
Antihypertensive medication ** (%)	38 (44.7)	28 (80.0)	6 (12.0)	**<0.0001**
Pregnancy weight gain (kg)	14.0 ± 7.1	13.2 ± 7.1	14.6 ± 7.1	0.393
Gestational age delivery	36.0 (32.2, 38.0)	37.5 (34.4, 39.4)	34.0 (31.0, 38.0)	**<0.001**
Delivery before 37 weeks gestation (%)	48 (56.5)	11 (31.4)	37 (74.0)	**<0.001**
Cesarean section (%)	44 (51.8)	14 (40.0)	30 (60.0)	0.081
Female infant (%)	42 (49.4)	13 (37.1)	29 (58.0)	0.078
Birth weight (g)	2200 (1495, 3098)	2655 (2075, 3280)	1920 (1285, 2351)	**0.0003**
Small for gestational age (<5th percentile)	15 (17.6)	5 (14.3)	10 (20.0)	0.573
Admission to NICU (%)	59 (69.4)	15 (42.9)	44 (88.0)	**<0.001**
Placental weight (g)	334 (274, 443)	382 (326, 516)	312 (236, 431)	0.057
At 6 months postpartum				
Systolic BP (mmHg)	119 ± 18	116 ± 23	121 ± 13	0.164
Diastolic BP (mmHg)	81 ± 10	78 ± 9	82 ± 10	0.081
Antihypertensive medication use (%)	13 (15.3)	5 (14.3)	8 (16.0)	1.00
Breastfeeding (%)	44 (52.4)	22 (64.7)	22 (44.0)	0.077
Total cholesterol	4.8 ± 1.0	4.9 ± 1.0	4.7 ± 1.0	0.292
Fasting glucose	4.8 ± 0.5	4.7 ± 0.5	4.8 ± 0.5	0.397
HDL	1.5 ± 0.4	1.5 ± 0.4	1.5 ± 0.4	0.541
LDL	2.8 (2.2, 3.4)	3.0 (2.2, 3.5)	2.6 (2.1, 3.3)	0.231
hsCRP	2.6 (1.0, 7.4)	2.6 (0.9, 8.4)	2.0 (0.98, 5.9)	0.443
Triglycerides	0.98 (0.67, 1.69)	0.98 (0.72, 1.88)	0.96 (0.65, 1.60)	0.500
Screen high-risk for lifetime CVD (%)	53 (62.4)	18 (51.4)	35 (70.0)	0.112

Data presented as mean ± SD, median (IQR) or *n* (%). BP: blood pressure; hsCRP: high sensitivity C-reactive protein; CVD: cardiovascular disease; HDL: high density lipoprotein; IQR: interquartile range; LDL: low density lipoprotein; NICU: neonatal intensive care unit; PE: preeclampsia; SD: standard deviation. **^§^** Comparison between participants in Ottawa and Kingston. * Highest BP measurement following admission before delivery. ** Medication started during index pregnancy or postpartum prior to discharge from hospital.

**Table 3 jcm-11-01576-t003:** Frequency of placental lesions by cardiovascular risk group.

Placental Lesion	High CVD Risk (*n* = 53)	Low CVD Risk (*n* = 32)	*p*-Value (Pearson *X*^2^)
Evidence of maternal vascular malperfusion			
Placental infarction	16 (30.2)	7 (21.9)	0.403
Distal villous hypoplasia	15 (28.3)	8 (25.0)	0.740
Accelerated villous maturation	31 (58.5)	12 (37.5)	0.061
Syncytial knots	34 (64.2)	17 (53.1)	0.315
Perivillous fibrin deposition	5 (9.4)	6 (18.8)	0.215
Villous agglutination	7 (13.2)	1 (3.1)	0.123
Presence of retroplacental hematoma	0 (0)	2 (6.3)	0.066
MVM Score of 0	8 (15.1)	9 (28.1)	0.146
MVM Score 2 or more	29 (54.7)	9 (28.1)	**0.017**
Evidence of maternal decidual arteriopathy			
Insufficient vessel remodeling	7 (13.2)	2 (6.3)	0.312
Fibrinoid necrosis	4 (7.5)	2 (6.3)	0.821
Decidual arteriopathy present	9 (17.0)	3 (10.3)	0.416
Evidence of ascending intrauterine infection			
Maternal inflammatory response	2 (3.8)	4 (4.7)	0.128
Fetal inflammatory response	2 (3.8)	2 (6.3)	0.601
Ascending intrauterine infection present	3 (5.7)	5 (15.6)	0.127
Evidence of placenta villous maldevelopment			
Chorangiosis	0 (0)	0 (0)	--
Chorangiomas	0 (0)	0 (0)	--
Delayed villous maturation	1 (1.9)	2 (6.3)	0.291
Evidence of fetal vascular malperfusion			
Avascular fibrotic villi	2 (3.8)	0 (0)	0.266
Thrombosis	1 (1.9)	1 (3.1)	0.715
Intramural fibrin deposition	0 (0)	3 (9.4)	**0.023**
Karyorrhexis	0 (0)	0 (0)	--
High-grade fetal vascular malperfusion	2 (3.8)	0 (0)	0.266
Fetal vascular malperfusion present	4 (7.5)	5 (15.6)	0.241
Fibrinoid			
Massive Perivillous fibrin deposition pattern	1 (1.9)	0 (0)	0.434
Maternal floor infarction pattern	0 (0)	0 (0)	--
Intervillous thrombi			
Intervillous thrombi	5 (9.4)	1 (3.1)	0.271
Evidence of chronic inflammation			
Villitis of unknown etiology	5 (9.4)	3 (9.4)	0.993
Chronic intervillositis	0 (0)	0 (0)	--
Chronic plasma cell deciduitis	5 (9.4)	3 (9.4)	0.993
Chronic inflammation present	7 (13.2)	6 (18.8)	0.492

Data presented as *n* (%). MVM: maternal vascular malperfusion.

## Data Availability

Reasonable requests for data will be considered.

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
