# Peer review of "Placental Pathology as a Tool to Identify Women for Postpartum Cardiovascular Risk Screening following Preeclampsia: A Preliminary Investigation"

_jcm, 2022, doi:10.3390/jcm11061576_

Round 1
Reviewer 1 Report
In this study the authors assess the utility of placental pathology for predicting those women with preeclampsia during pregnancy who will screen at high risk for cardiovascular disease 6 months after delivery. This study addresses a significant need—the identification of those women who will go on to develop early cardiovascular disease. Given that a number of previous studies have provided some evidence for the connection between placental pathology and later life cardiovascular disease risk, the findings are unsurprising if modest. As the authors themselves point out, a larger study is clearly needed to better characterize these relationships. That said, this small but careful study is a useful addition to this body of evidence.
Characteristics at the time of delivery vary quite substantially between the two centres, with the Kingston cohort substantially more severely affected (BP, PTB, birthweight, NICU admission). Why are these two groups so markedly different? If, as suggested in the Strengths and Limitations section, this is due to differences in the submissions of placentas to Pathology, could that be investigated? The mere presence of these unexplained major differences raises questions about the validity of this study.
What is the estimated cardiovascular disease risk for this cohort if one looks only at clinical factors (maternal age, gestational weight gain, etc.)? In other words, does placental pathology add anything that wouldn’t already have been identified based on clinical factors alone? This seems a critical issue in assessing the utility of placental pathology (which is an added cost) for initiating cardiovascular risk screening.
Chronic utero-placental separation (an Amsterdam criterion for MVM) is listed as one of the placental pathology categories in the Methods section, but it does not appear in the Results section (Table 3). Were there no such cases?
What is the difference between “perivillous fibrin deposition” in the MVM category and the “perivillous fibrin deposition” in the entire “fibrinoid” section?
Reviewer 2 Report
Thank you for the opportunity to review this interesting work. I have the following comments for the authors' consideration:
Introduction: Please add a recently reference, for example reference number 1 it was from 2010, 12 years almost, need more recently references about PE prevalence. Similarly, for reference number 2 it’s since 2007.
Introduction: line51-52 Proposed mechanisms of placenta lesion involvement in PE and CVD later in life are not clearly described need more clarification. Then, please provide a strong reference supporting this mechanism.
Methods: Please provide a sub-section for materials and methods section, it’s not clear and a lot of information which we cannot follow …, i.e add a sub-section:
-Human cohort:
- Ottawa
- Kingston
-Sample collection and preparation:
- Ottawa
- Kingston
-Statistical Analysis: please add: Statistical significance was set at p ????
Methods: Line 149: “A random urine sample was collected and the albumin: creatinine ratio was measured.” I ‘am not sure the authors used this measurement for analyses in this article ?? clarify please?
Methods: Are there any concerns about use of the small cohort and between Kingston, n= 50; and Ottawa, n=35? under/powered to find an association (statically representative cohort) between placental pathology and postpartum CVD risk in women following PE?
Results: Line 181: I'm not sure I understand what the authors mean, it’s well known that age is an important risk factor for PE and CVD!!
Results: It is necessary to unify the typeface and writing style of each table, whether items with significant differences are bolded or marked, p value statically significant fixed at?? (ie Data with significant differences are in bold.)". Usually we add “Data Presented as mean ± SD, median [IQR] or n (%)” below the table. This comment for all the tables, please modifies. Table 3: “Delivery and postpartum characteristics of the study participants as a combined cohort and by individual study site”. Data presented were from combined cohort but not shown by individual site?? Why? It’s important to add this result by individual study site in the table and see the difference between Kingston and Ottawa. I would like to see if after adjusting for pre-pregnancy BMI and age, the difference between the sites. Indeed, BMI and age are important risk factors for pregnancy complications (PE) and CVD later in life.
Discussion: I think that the background literature concerning PE and placental dysfunction (line 291), need to include also the implication of renin-angiotensin system (RAS) play an important role in placental dysfunction, which a local RAS has been described and implicated in placenta development and may be involved in the CVD.
Author Response
Please see the attachment (reviewer 2).

Reviewer 3 Report
This study investigates the association between placental histopathological lesions and the assessment of cardiovascular risk at 6 months postpartum in women diagnosed with preeclampsia.
The study is well-written. The methodology and results are scientifically sound. The limitations of the study are accurately discussed.
I have just a question: Were there any differences in placental lesions between the two study sites? Or between preterm and term cases of preeclampsia? Since it is known that preterm cases of preeclampsia are usually more severe and associated with more severe placental lesions. Moreover, it seems that included cases from Kingston are mostly severe and preterm cases.
Author Response
Reviewer #3, Comment #1: Were there any differences in placental lesions between the two study sites? Or between preterm and term cases of preeclampsia? Since it is known that preterm cases of preeclampsia are usually more severe and associated with more severe placental lesions. Moreover, it seems that included cases from Kingston are mostly severe and preterm cases.
Response: We thank this reviewer for these questions. We have looked at the frequencies of placental lesions by CVD risk group stratified by individual site and found no significant differences (please see response to Reviewer #2, Comment #9). We have added a sentence to the Results section on Line 183-184, page 4: “There were no differences in the frequencies of placental lesions between high and low-risk groups when stratified by individual study site (data not shown).” We could expect that there would be more placental pathology in the cases from Kingston given their pregnancy characteristics (ex: earlier GA at delivery), however, as we have not find differences in placental lesions between the sites, we believe this is reflecting the heterogeneity in placental disease across the spectrum of preeclampsia. In fact, an additional regression analysis using gestational age at delivery showed a non-significant OR of 0.90 (0.80-1.02). In contrast, MVM lesions with a severity score >2 gave a significant OR of 3.10 (1.20 -7.92), indicating that placental pathology better identifies CVD risk than gestational age at delivery alone.
Round 2
Reviewer 1 Report
I have looked over the revisions, and the authors need one more tiny but important revision. Specifically, in reference to Comment #4, the pathologic entity "laminar necrosis of the decidua capsularis" is not a feature of fetal vascular malperfusion. It has been suggested as a feature of maternal vascular malperfusion, but the limited studies on this entity are contradictory.
Author Response
We greatly appreciate your feedback. Upon reviewing the literature surrounding the classification of this lesion, we agree that its inclusion in the fetal vascular malperfusion category is not supported. Further, it appears there in insufficient evidence to place this lesion in any specific placenta pathology group at this time due to lack of clarity in the literature and in the Amsterdam consensus regarding its diagnostic criteria, etiology/pathophysiology and clinical significance. As such, we have chosen to remove laminar necrosis of the decidua capsularis from our analysis given that this lesion was only observed in a small number of cases (n=4) and that its inclusion did not result in any significant changes to our results (provided below). The appropriate changes have been made to the FVM category in Table 3 of the manuscript.
Results including laminar necrosis of the decidua capsularis in MVM category:
Unadjusted odds ratio for MVM score >2: 3.1 ([1.2, 7.9], p= 0.019)
Adjusted odds ratio for MVM score >2: 3.6 ([1.1, 9.3], p= 0.027)
Area under the curve for predictive model with MVM score >2 (Figure 1b): 0.63 [0.51, 0.75]
Area under the curve for predictive model with MVM score >2 with clinical variables (Figure 1c): 0.72 [0.61, 0.83]
Reviewer 2 Report
no comments to add!
Author Response
Thank you.